# BIOREASON: Incentivizing Multimodal Biological Reasoning within a DNA-LLM Model

**Adibvafa Fallahpour**[*,1,2,3,5]
adibvafa.fallahpour@mail.utoronto.ca

**Andrew Magnuson**[*,1,2]
andrew.magnuson@mail.utoronto.ca

**Purav Gupta**[*,1,2]
purav.gupta@mail.utoronto.ca

**Shihao Ma**[1,2,3]
shihao.ma@mail.utoronto.ca

**Jack Naimer**[1,2,3]
jack.naimer@mail.utoronto.ca

**Arnav Shah**[1,2,3]
arnav.shah@mail.utoronto.ca

**Haonan Duan**[1,2]
haonan.duan@mail.utoronto.ca

**Omar Ibrahim**[3]
omar.ibrahim2@uhn.ca

**Hani Goodarzi**[†,4,6]
hani.goodarzi@ucsf.edu

**Chris J. Maddison**[†,1,2,7]
cmaddis@cs.toronto.edu

**Bo Wang**[†,1,2,3]
bowang@vectorinstitute.ai

[1]University of Toronto   [2]Vector Institute   [3]University Health Network (UHN)
[4]Arc Institute   [5]Cohere   [6]University of California, San Francisco   [7]Google DeepMind

## Abstract

Unlocking deep, interpretable biological reasoning from complex genomic data is a major AI challenge hindering scientific discovery. Current DNA foundation models, despite strong sequence representation, struggle with multi-step reasoning and lack inherent transparent, biologically intuitive explanations. We introduce BIOREASON, a pioneering architecture that, for the first time, deeply integrates a DNA foundation model with a large language model (LLM). This novel connection enables the LLM to directly process and reason with genomic information as a fundamental input, fostering a new form of multimodal biological understanding. BIOREASON's sophisticated multi-step reasoning is developed through supervised fine-tuning and targeted reinforcement learning, guiding the system to generate logical, biologically coherent deductions. Across biological reasoning benchmarks, BIOREASON significantly improves performance, raising accuracy on KEGG-based disease pathway prediction from 86% to 98% and delivering an average 15% gain over strong single-modality baselines in variant effect prediction tasks. BIOREASON reasons over unseen biological entities and articulates decision-making through interpretable, step-by-step biological traces, offering a transformative approach for AI in biology that enables deeper mechanistic insights and accelerates testable hypothesis generation from genomic data. Data, code, and checkpoints are publicly available at https://github.com/bowang-lab/BioReason.

---

*Equal contribution.        †Equal advising.

39th Conference on Neural Information Processing Systems (NeurIPS 2025).

# 1 Introduction

Biological data, spanning genomics, transcriptomics, biomedical literature, and more, is expanding at an unprecedented rate, creating immense opportunities for scientific discovery. This data explosion has catalyzed the development of foundation models (FMs), deep networks trained on vast datasets that enable a wide array of downstream tasks. In genomics, DNA foundation models [6, 10, 31, 45, 13] have demonstrated remarkable capabilities by learning dense sequence representations that drive splice site identification, variant effect prediction, and regulatory element characterization.

Despite these advances, a critical challenge with foundation models still persists: effectively translating these learned representations into mechanistic insights and falsifiable hypotheses. Current DNA foundation models, while powerful in their representational capacity, typically function as "black boxes" that lack the inherent ability to generate transparent, biologically intuitive explanations [4, 26]. These limitations are prominent in complex biological problems requiring mechanistic understanding, such as gene pathway analysis, phenotype prediction, and disease mechanism elucidation [9].

Large language models (LLMs) [32, 2, 12, 34] have rapidly advanced in reasoning capabilities, problem-solving, and knowledge depth. Through sophisticated training methods including reinforcement learning and supervised fine-tuning, these models demonstrate increasingly sophisticated multi-step reasoning across domains from mathematical problem-solving to logical deduction [14, 28, 30, 18]. However, LLMs alone lack the specialized architecture to effectively process raw genomic sequences and often fail to capture nuanced biological patterns in genetic data.

This disconnect between powerful sequence representations of DNA foundation models and sophisticated reasoning capabilities of LLMs creates a significant barrier to developing AI systems that provide deep mechanistic insights comparable to biology domain experts. To bridge this gap, we present BIOREASON: a novel architecture that fundamentally integrates a DNA foundation model with an LLM, enabling a new paradigm of multimodal biological understanding and reasoning.

BIOREASON is distinguished by its ability to create a unique flow of information between genomic and natural language. This architecture enables the system to process raw DNA sequences while leveraging the reasoning capabilities of modern LLMs to generate biologically coherent explanations and predictions. Through a training methodology combining supervised fine-tuning and reinforcement learning, BIOREASON develops the capacity for sophisticated multi-step reasoning over genomic data; a capability that neither DNA foundation models nor LLMs can achieve independently.

**Contributions.** Our key contributions include:

- **Novel multimodal architecture.** The first successful integration of a large DNA foundation model with an LLM, establishing a new methodology for AI-driven biological studies.

- **Advanced reasoning.** A systematic training approach combining supervised fine-tuning and reinforcement learning that incentivizes multi-step biological reasoning.

- **New biological reasoning benchmarks.** Development and curation of novel benchmarks for evaluating biological reasoning capabilities, including an annotated reasoning dataset for gene pathway and disease prediction dataset from KEGG [20].

- **Empirical performance improvements.** Demonstration that BIOREASON outperforms both DNA foundation models and LLMs with average performance gains of 15%+ over baseline.

- **Interpretable reasoning traces.** A mechanism for generating step-by-step biological reasoning traces that provide interpretable predictions, enhancing scientific insight and hypothesis generation

# 2 Background & Related Work

## 2.1 DNA Foundation Models

Recent years have witnessed the emergence of DNA foundation models [6, 10, 43, 31] that have significantly accelerated discovery throughout the biological sciences. These models extract meaningful representations directly from nucleotide sequences by pre-training on vast genomic datasets.

Moreover, comprehensive benchmarking studies [15] have demonstrated their proficiency across various genomics tasks in both zero-shot and fine-tuned settings.

Evo2 [6], in particular, represents a significant advancement as one of the largest genomic foundation models to date, enabling extremely long-range context windows and predictions. Its ability to generate complete bacterial and yeast genomes underscores the potential of these models to capture complex genomic patterns [22]. However, a critical limitation persists: these foundation models operate as "black boxes," lacking the interpretability necessary to explain how they derive conclusions from their embeddings. This opacity hampers the advancement of biological knowledge by obscuring the mechanistic insights that could otherwise be derived from model predictions.

## 2.2 Large Language Models for Biological Reasoning

LLMs have demonstrated remarkable capabilities in understanding and generating human-like text, with substantial success in interpreting and reasoning over complex biomedical data. Recent reviews [43] highlight their success across diverse domains, from clinical applications involving patient notes to biological research contexts. The development of specialized models pre-trained on biomedical literature [25], has further enhanced their domain-specific performance.

Genomics-focused LLMs such as GeneGPT [19], agentic models such as TxGemma [40], and scientific reasoning models such as rbio1 [18] represent initial attempts to integrate language models with genomic databases. ChatNT [35] takes a step further by integrating DNA foundation model representations with language models in a multimodal framework. However, no previous work has trained such systems for complex biological reasoning. We present the first multimodal framework designed and trained to perform biological reasoning by combining textual knowledge from LLMs with nucleotide-level representations from DNA foundation models.

## 2.3 Genomics Benchmarks

DNA foundation models are typically evaluated on established benchmarks encompassing diverse prediction tasks, including regulatory element identification, variant effect prediction, transcription factor binding site prediction, and splice site classification. Comprehensive benchmarking frameworks like BEND [29] provide standardized evaluation protocols that enable meaningful comparisons between models across these supervised tasks.

While these benchmarks effectively measure performance on specific downstream applications, they inadequately evaluate a model's capacity for higher-order reasoning or hypothesis generation, capabilities essential for advancing scientific understanding. This represents a critical conceptual gap between current evaluation metrics and the sophisticated reasoning abilities desired from next-generation foundation models. The field requires benchmarks that challenge models to perform multi-step logical reasoning and predict potential biological mechanisms. This need motivated our curation of the KEGG pathway database [20] to create a multi-step reasoning, variant effect prediction dataset that specifically evaluates a model's capacity for mechanistic biological reasoning.

## 3 BioReason Model

We introduce BIOREASON, a multimodal framework designed to unlock deep, interpretable biological reasoning by synergistically integrating genomic and language data. BIOREASON operates on two primary input streams: (i) one or more genomic sequences, denoted $S_{\text{DNA}}$; and (ii) textual queries, $Q_{\text{TEXT}}$. These queries are processed by an LLM-specific tokenizer, $T_{\text{LLM}}(\cdot)$, into a sequence of $M$ tokens $(w_1, \ldots, w_M)$ from the LLM's vocabulary $\mathcal{V}_{\text{LLM}}$. Current methods often fall short in this domain: LLMs treat $S_{\text{DNA}}$ as simple strings, thereby missing rich genomic features, while DNA Foundation Models ($f_{\text{DNA}}$) capture these features but primarily yield task-specific discriminative outputs (e.g., classification or regression scores) rather than interpretable natural language. BIOREASON bridges this gap by deriving contextualized DNA embeddings from the $S_{\text{DNA}}$ input(s) and integrating them with the tokenized $Q_{\text{TEXT}}$ to form a unified multimodal input sequence, $X_{\text{LLM}}$, for its core LLM. This direct integration enables the generation of explanatory text, $Y_{\text{OUT}} = (y_1, \ldots, y_K)$, grounded in genomic nuances. The output $Y_{\text{OUT}}$ presents biological reasoning and the final response. Figure 1 depicts the overall architecture.

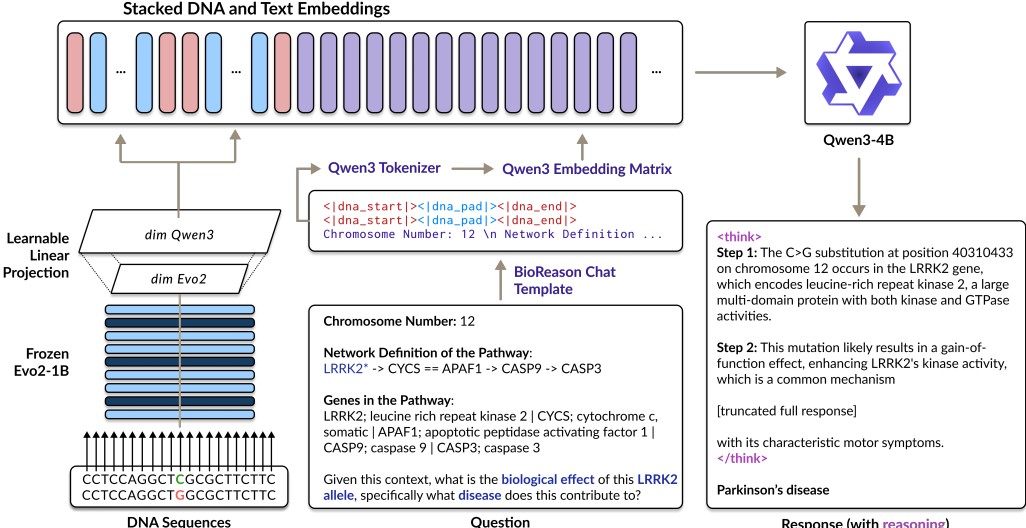

Figure 1: **BIOREASON Architecture.** Schematic representation of our novel multimodal framework that integrates a DNA foundation model with a Large Language Model.

## 3.1 DNA Foundation Model ($f_{\mathrm{DNA}}$) Encoder

$f_{\mathrm{DNA}}$ transforms each input $S_{\mathrm{DNA}}$ sequence into contextualized embeddings. We utilize established DNA foundation models such as StripedHyena2 (e.g., Evo2) [6, 33], or the Nucleotide Transformer (NT), [11], as the $f_{\mathrm{DNA}}$. Each $S_{\mathrm{DNA}}$ is first processed by its respective DNA-specific tokenizer, $T_{\mathrm{DNA}}(\cdot)$, which segments it into a sequence of $L'$ DNA tokens, $D = (d_1, \ldots, d_{L'})$; each token $d_j$ can represent one or more nucleotides. If an input $S_{\mathrm{DNA}}$ sequence, after tokenization by $T_{\mathrm{DNA}}$, exceeds a defined context length (e.g., 2048 DNA tokens), it is truncated. The chosen $f_{\mathrm{DNA}}$ architecture then maps each token sequence $D$ to a sequence of high-dimensional per-token embeddings $E_{\mathrm{DNA}} = (e_1, \ldots, e_{L'}) \in \mathbb{R}^{L' \times d_{\mathrm{dna}}}$. These $d_{\mathrm{dna}}$-dimensional embeddings capture context-dependent genomic features. The weights of the $f_{\mathrm{DNA}}$ are kept frozen during BIOREASON's training and inference.

## 3.2 Large Language Model ($f_{\mathrm{LLM}}$) Backbone

The $f_{\mathrm{LLM}}$ is the primary reasoning engine and text generator. We employ Qwen3 [41, 42], an autoregressive Transformer-based LLM, initialized with its original pre-trained weights. This model receives the multimodal input sequence $X_{\mathrm{LLM}}$ and is trained to predict the next token $y_i$ in the sequence $Y_{\mathrm{OUT}}$, conditioned on the preceding tokens $y_{<i}$ and $X_{\mathrm{LLM}}$. Mathematically, we optimize the parameters $\theta_{\mathrm{LLM}}$ of the $f_{\mathrm{LLM}}$ by maximizing the log-likelihood of the observed sequences:

$$\mathcal{L} = \sum_i \log P(y_i | y_{<i}, X_{\mathrm{LLM}}; \theta_{\mathrm{LLM}}) \tag{1}$$

The $f_{\mathrm{LLM}}$ utilizes special tokens to structure conversational interactions and reasoning within its textual output $Y_{\mathrm{OUT}}$. These include tokens defining user and assistant roles (e.g., `<|im_start|>user/assistant ...<|im_end|>`), structuring reasoning steps (e.g., `<think>...</think>`), alongside standard padding tokens (`<|endoftext|>`).

## 3.3 Multimodal Genomic Integration

Genomic information, as DNA embeddings from $f_{\mathrm{DNA}}$, is integrated into the $f_{\mathrm{LLM}}$'s input by stacking these with embeddings of the user's query $Q_{\mathrm{TEXT}}$ and special tokens such as `<dna_start>` and `<dna_end>`. Key to this integration is the preparation of the DNA embedding block, $\mathbf{E}'_{\mathrm{DNA}}$, formed from one or more input DNA sequences. For each sequence $S_{\mathrm{DNA},k}$, its $f_{\mathrm{DNA}}$-generated embedding sequence $E_{\mathrm{DNA},k} \in \mathbb{R}^{L'_k \times d_{\mathrm{dna}}}$ (where $L'_k$ is its tokenized length) is projected by a learnable linear layer, $\mathrm{Proj} : \mathbb{R}^{d_{\mathrm{dna}}} \to \mathbb{R}^{d_{\mathrm{llm}}}$, to yield $\mathbf{E}'_{\mathrm{DNA},k}$ of dimension $d_{\mathrm{llm}}$. The resulting $\mathbf{E}'_{\mathrm{DNA}}$ block is obtained by stacking all $\mathbf{E}'_{\mathrm{DNA},k}$ sequences along the sequence dimension.

Concurrently, the user's tokenized query $Q_{\text{TEXT}} = (w_1, \ldots, w_M)$ is mapped to its embedding sequence $\mathbf{E}_{Q_{\text{text}}} = (E(w_1), \ldots, E(w_M))$ by the $f_{\text{LLM}}$'s input embedding layer, $E(\cdot)$. Similarly, the special tokens `<dna_start>` and `<dna_end>` are embedded via $E(\cdot)$ to produce $e_{\texttt{<dna\_start>}}$ and $e_{\texttt{<dna\_end>}}$. These components are then stacked along the sequence dimension to form the multimodal input $X_{\text{LLM}}$ for $f_{\text{LLM}}$.

$$X_{\text{LLM}} = (e_{\texttt{<dna\_start>}}, \mathbf{E}'_{\text{DNA}}, e_{\texttt{<dna\_end>}}, \mathbf{E}_{Q_{\text{text}}}) \tag{2}$$

All constituent embedding vectors within $X_{\text{LLM}}$ receive positional information via Rotary Position Embedding (RoPE) [39], applied according to their final sequence positions. This strategy enables $f_{\text{LLM}}$ fine-grained attention over both genomic and textual components within a unified modality.

### 3.4 Group Relative Policy Optimization (GRPO)

To further enhance BIOREASON 's reasoning performance beyond supervised fine-tuning, we employ Group Relative Policy Optimization (GRPO) [36, 12], a reinforcement learning strategy tailored for refining reasoning generation in language models. GRPO leverages reward signals within groups of sampled outputs, eliminating the need for an explicit value estimator. We implement Dr. GRPO [27], an unbiased variant of GRPO that improves token efficiency while maintaining performance.

For the full formalism, including the composite reward design, advantage normalization, and the clipped surrogate objective with KL regularization, please refer to Appendix A.4.

## 4 Datasets

To develop a multimodal DNA-LLM model with reasoning capabilities, we curated three datasets: one novel dataset specifically designed to incentivize reasoning and two adapted from established benchmarks. The adapted datasets are derived from ClinVar [24] and OMIM [1], which are widely used for variant effect prediction tasks. Our novel dataset is based on KEGG Network Variants data [20] and enhanced with cross-linked metadata from several public variant repositories including ClinVar [24], OMIM [1], dbSNP [37], and COSMIC [38]. This novel dataset relies on the high-quality manual annotations and descriptions from the curators of KEGG, for gene pathway descriptions and downstream phenotypic effects, like disease.

### 4.1 KEGG-Derived Biological Reasoning Dataset

#### 4.1.1 Dataset Integration and Statistics

We present a high-quality biological reasoning dataset derived from the Kyoto Encyclopedia of Genes and Genomes (KEGG) pathway database [20], consisting of 1,449 entries that elucidate the mechanistic connections between genetic variants and disease phenotypes. As seen in Figure 2, the dataset construction involved a rigorous multi-stage process that integrates structured pathway information with variant data to enable step-by-step reasoning across molecular networks.

For primary data integration, we extracted pathway network data from KEGG [20], focusing on disease-associated molecular interactions. Pathway data was augmented with variant information from clinical databases (ClinVar, dbSNP, OMIM, COSM) [24, 37, 1, 38] through a semi-automated mapping protocol [17, 21] that preserved relational integrity between genomic loci and functional elements within pathways. Each molecular network was represented using a standardized symbolic notation (e.g., "GENE1+GENE2 -> GENE3 -| GENE4") that encapsulates interaction types including activation, inhibition, complex formation, and transcriptional regulation.

To support variant interpretation, we included paired reference and variant sequences with precise alignment coordinates. These sequences have an average length of approximately 4,000 base pairs, with most variants differing by only 1–3 nucleotides from their reference sequences.

#### 4.1.2 Reasoning Path Construction and Curation

A distinctive feature of this dataset is its inclusion of explicit causal reasoning paths connecting genetic variants to disease phenotypes via defined molecular mechanisms; these paths were constructed

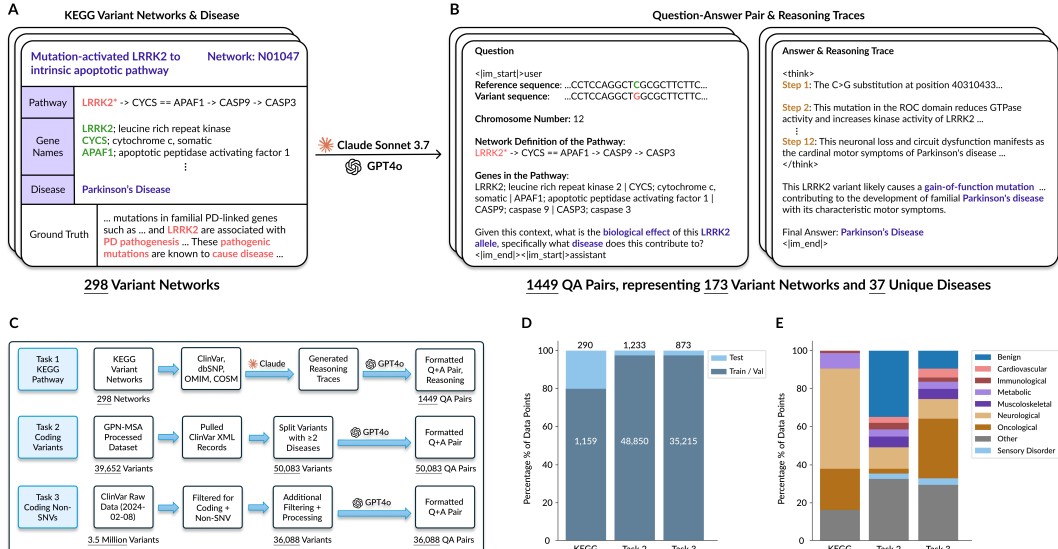

Figure 2: **BIOREASON Dataset Curation and Composition. A.** Representative example of a KEGG Variant Network element from the 298 networks utilized in our study, illustrating the relationship between genomic variants and their corresponding disease annotation that serves as ground truth for generating mechanistic reasoning traces. **B.** Exemplar of a structured question-answer pair with an accompanying multi-step reasoning trace demonstrating the expected logical progression from genomic variant to phenotypic outcome. **C.** Pipeline for data acquisition, integration, and curation across the three BIOREASON tasks. **D.** Distribution of train/test splits across the three curated datasets. 10% of train dataset was used for validation. **E.** Distribution of disease categories represented within the datasets, highlighting the diversity of variants and diseases represented in the datasets.

using the Claude 3.7 Sonnet model [2] and grounded with contextual disease information from the KEGG disease database [20]. For training and evaluation, the dataset is structured into standardized question-answer pairs: questions (illustrated in Figure 2B) incorporate variant details, network definitions, and gene descriptions, while answers provide concise mechanism-disease associations. The accompanying reasoning paths (mean length: 303.8 words) elaborate these mechanistic variant-to-phenotype links with precise molecular information.

### 4.2 Variant Effect Prediction of Coding Sequences

This dataset originated from the GPN-MSA [5] study. Affected gene names and disease phenotypes were extracted from ClinVar [24] XML records (via NCBI's Entrez Direct tool [21]), while benign variants were sourced from gnomAD v3.1.2 [8] (requiring allele number $\geq 25,000$ and minor allele frequency (MAF) > 5%). The data was split by chromosome (Chr 1–7, 9–22, X, Y for training; Chr 8 for testing). For training augmentation, GPT-4o [32] generated 50 semantically equivalent question variations per sample, prompting for pathogenic/benign classification and conditional disease phenotype prediction using chromosome and gene context; mutations linked to multiple diseases were treated as distinct samples for comprehensive phenotype coverage.

### 4.3 Variant Effect Prediction of Coding Non-SNVs

Coding non-SNVs were sourced from the ClinVar [24] database (2024-02-28 release). We filtered variants to retain only coding non-SNVs within the nuclear genome, affecting $\leq 64$ base pairs, of certain significance, and with a review status of at least two stars, matched to GRCh38.p14 transcripts. After extracting affected gene names and disease phenotypes where available, a custom algorithm partitioned the dataset to ensure balanced disease representation in train/test splits. Finally, to augment training data, GPT-4o [32] generated 50 semantically equivalent question variations for each entry, using gene and chromosome number as context, prompting for pathogenic/benign classification and, if pathogenic, the associated disease phenotype.

# 5 Experiments

## 5.1 Datasets

BIOREASON's performance is evaluated on three datasets (detailed in Section 4):

**KEGG-Derived Biological Reasoning Dataset.** This dataset (1,449 variants, 37 unique diseases) evaluates multi-step mechanistic reasoning. Input: paired reference and variant DNA sequences ($S_{DNA}$), and a textual query ($Q_{TEXT}$) with pathway/gene context. Task: predict the mutation's effect and resulting disease by sequence-to-sequence generation of $Y_{OUT}$ containing step-by-step reasoning between `<think>` (special tokens) and the final disease.

**Variant Effect Prediction of Coding Sequences (VEP-Coding).** Comprising 50,083 core variant entries, this dataset tests classifying coding variants. Input: paired reference and variant DNA sequences ($S_{DNA}$), and a textual query ($Q_{TEXT}$) providing gene and chromosome context. Task: sequence generation to predict if a variant is benign, or pathogenic with its associated disease. Split: Chromosomes (Chr) 1–7, 9–22, X, Y for train/validation; Chr 8 for testing.

**Variant Effect Prediction of Coding Non-SNVs (VEP-Non-SNV).** Containing 36,088 core non-SNV entries, this dataset addresses non-SNV alterations (e.g., indels <64 bp). Input: paired reference and variant DNA sequences ($S_{DNA}$), and an augmented textual query ($Q_{TEXT}$) providing gene and chromosome context. Task: sequence generation to predict if a non-SNV is benign, or pathogenic with its associated disease(s). We used stratified train/test splits to ensure balanced disease representation.

## 5.2 Models and Baselines

To benchmark BIOREASON's performance, we evaluated it against several baseline models, categorized as DNA foundation models ($f_{DNA}$) and Large Language Models ($f_{LLM}$).

For $f_{DNA}$ baselines, we utilized pre-trained Evo2-1B [6] and Nucleotide Transformer (NT-500M) [11] models. For downstream predictions, $f_{DNA}$ models were adapted with an attention head where a single learnable query vector attends to the sequence token embeddings to produce a final sequence representation. For ($f_{LLM}$) baselines, we fine-tuned pre-trained Qwen3 models of two sizes: Qwen3-1.7B and Qwen3-4B [41, 42, 34]. These models were trained to receive text queries and DNA sequences treated as plain text strings and generate text with reasoning steps and final predictions.

The proposed BIOREASON models, were evaluated in several $f_{DNA}$ and $f_{LLM}$ combinations. Specifically, we tested Evo2-1B and NT-500M as $f_{DNA}$ encoders, each paired with Qwen3-1.7B and Qwen3-4B as $f_{LLM}$ backbones. The primary training methodology for all BIOREASON configurations was Supervised Fine-Tuning (SFT). Reinforcement Learning (RL) fine-tuning using the GRPO algorithm was subsequently applied to select DNA-LLM models.

## 5.3 Experimental Setup

Our experimental setup varied by model architecture—BIOREASON, LLM-only, or $f_{DNA}$-only—and task. BIOREASON and LLM-only models underwent Supervised Fine-Tuning (SFT), with LLM parameters efficiently updated via Low-Rank Adaptation (LoRA) [16]. For $f_{DNA}$-only baselines, core DNA model weights were frozen; only a task-specific attention head and classifier were trained.

SFT objectives for these models differed: for the KEGG Dataset Task, models generated reasoning steps between `<think>` tokens and a final disease prediction. For VEP Datasets Tasks, they aimed for pathogenic/benign classification and conditional disease prediction for pathogenic variants. During SFT, a specialized attention mask restricted loss computation exclusively to the response between `<think>` tokens and final answer tokens, excluding those from the input query or DNA embeddings. Select BIOREASON models were further optimized with GRPO. Details for LoRA configurations, all SFT and RL hyperparameters, and GRPO reward functions are provided in Appendix A.1.

Performance evaluation metrics were task-specific. The KEGG Dataset Task utilized Accuracy, Macro F1-score, Macro Precision, and Macro Recall as a multi-class disease prediction assessment, considering potential class imbalances. For VEP Datasets Tasks, Accuracy and F1-score measured the binary pathogenic/benign classification. All LLM and DNA-LLM generations were deterministic with a decoding temperature of 0. We leveraged vLLM for fast inference. [23]

Table 1: Performance comparison of $f_{\text{DNA}}$-only, LLM-only, and DNA-LLM (BIOREASON) models on 290 test datapoints of the KEGG-derived biological reasoning task.

| Model | Accuracy | F1-Score | Precision | Recall |
|---|---|---|---|---|
| **[DNA] NT - 500M** | 86.55 | 69.76 | 73.23 | 66.61 |
| **[DNA] Evo2 - 1B** | 88.28 | 72.43 | 75.23 | 69.83 |
| **[LLM] Qwen3 - 1B** | 85.17 | 65.71 | 71.39 | 64.19 |
| **[LLM] Qwen3 - 4B** | 90.00 | 79.66 | 88.24 | 75.08 |
| **[DNA-LLM] NT + Qwen3 - 1B** | 89.31 | 81.46 | 88.24 | 77.30 |
| **[DNA-LLM] NT + Qwen3 - 1B (+GRPO)** | 91.72 | 75.06 | 79.41 | 72.89 |
| **[DNA-LLM] NT + Qwen3 - 4B** | 95.86 | 86.25 | 88.24 | 84.95 |
| **[DNA-LLM] NT + Qwen3 - 4B (+GRPO)** | **98.28** | 90.15 | 91.18 | 89.62 |
| **[DNA-LLM] Evo2 + Qwen3 - 1B** | 90.42 | 75.62 | 77.42 | 73.91 |
| **[DNA-LLM] Evo2 + Qwen3 - 4B** | 95.17 | 86.14 | 91.18 | 83.33 |
| **[DNA-LLM] Evo2 + Qwen3 - 4B (+GRPO)** | **98.28** | **93.05** | **94.12** | **92.48** |

Table 2: Performance comparison of $f_{\text{DNA}}$-only, LLM-only, and DNA-LLM (BIOREASON) models on Variant Effect Prediction (VEP) benchmarks (VE-Coding with 1.23K and VE-Non-SNV with 873 test datapoints), evaluating pathogenic/benign classification.

| Model | Variant Effect - Coding | | Variant Effect - Non-SNV | |
|---|---|---|---|---|
| | Accuracy | F1-Score | Accuracy | F1-Score |
| **[DNA] NT - 500M** | 60.91 | 45.20 | 67.93 | 65.97 |
| **[DNA] Evo2 - 1B** | 70.07 | 49.19 | 76.17 | 66.51 |
| **[LLM] Qwen3 - 1B** | 46.55 | 34.82 | 70.67 | 76.21 |
| **[LLM] Qwen3 - 4B** | 48.99 | 39.58 | 61.86 | 67.60 |
| **[DNA-LLM] NT + Qwen3 - 1B** | 55.58 | 54.50 | 72.82 | 76.93 |
| **[DNA-LLM] NT + Qwen3 - 4B** | 60.94 | 55.66 | 65.59 | 73.00 |
| **[DNA-LLM] Evo2 + Qwen3 - 1B** | 72.83 | 68.90 | **88.20** | **89.91** |
| **[DNA-LLM] Evo2 + Qwen3 - 4B** | **80.21** | **80.00** | 83.85 | 85.02 |

Beyond variant-effect tasks, we also experimented with BIOREASON using the supervised DNA foundation model Enformer [3] for chromatin accessibility prediction, see Appendix B.

## 5.4 Quantitative Results

BIOREASON's DNA–LLM hybrids deliver consistent, substantial gains over single-modality baselines on the KEGG-derived reasoning benchmark (Table 1). The Evo2+Qwen3-4B model with GRPO reaches 98.28% accuracy and 93.05% F1, outperforming the standalone Qwen3-4B (90.00%/79.66%) and Evo2 DNA-only (88.28%/72.43%) models. Notably, scaling from Qwen3-1B to Qwen3-4B substantially amplifies both base performance and GRPO effectiveness: while the 1B backbone shows mixed results with GRPO (accuracy improves from 89.31% to 91.72%, but F1 actually declines from 81.46% to 75.06%), the 4B backbone demonstrates dramatic and consistent improvement with GRPO, jumping from 95.17% to 98.28% accuracy and from 86.14% to 93.05% F1. This pattern holds across DNA foundation models; NT+Qwen3-4B with GRPO also reaches 98.28% accuracy with 90.15% F1, confirming that larger LLM backbones provide a significantly more effective substrate for reinforcement learning refinement while simultaneously elevating overall hybrid performance.

On the VEP benchmarks (Table 2), the DNA-LLM hybrids maintain their advantage across variant effect prediction tasks. Evo2+Qwen3-4B achieves 80.21% accuracy and 80.00% F1 in coding variant classification, far surpassing DNA-only (70.07%/49.19%) and LLM-only (48.99%/39.58%) baselines. For non-SNV classification, Evo2+Qwen3-1B leads with 88.20% accuracy and 89.91% F1, surpassing DNA-only (76.17%/66.51%) and LLM-only (70.67%/76.21%). Class-wise breakdowns for the KEGG benchmark are reported in Appendix C.

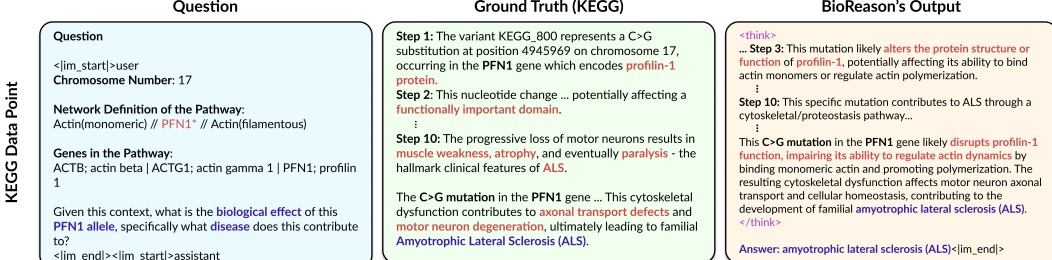

Figure 3: **Case Study of BIOREASON's Output**

## 5.5 Case Study

To illustrate BIOREASON's reasoning capabilities, consider its analysis of a PFN1 allele on chromosome 17 within the pathway `Actin(monomeric) // PFN1* // Actin(filamentous)`. BIORE-ASON correctly predicted Amyotrophic Lateral Sclerosis (ALS) as the resultant disease. Significantly, the model generated a plausible 10-step mechanistic rationale, initiating by identifying a specific C>G substitution in the PFN1 gene. Its reasoning then connected this variant to profilin-1 dysfunction, impaired actin dynamics critical for cytoskeletal integrity, subsequent disruption of axonal transport in motor neurons, and finally, the motor neuron degeneration characteristic of ALS. This example highlights BIOREASON's ability to not only make accurate predictions but also to articulate a step-by-step, biologically coherent pathway from a genomic variant to a complex disease phenotype.

## 6 Discussion

BIOREASON successfully integrates DNA foundation models with large language models, enabling direct LLM reasoning on genomic data. This overcomes key limitations of opaque DNA models and the inability of LLMs to natively process DNA sequences, resulting in enhanced multi-step biological reasoning and superior predictive performance over single-modality approaches.

A core strength of BIOREASON is its interpretable reasoning. By processing contextualized DNA embeddings within the LLM, cultivated through supervised fine-tuning, the system provides not only accurate predictions but also articulates its decision-making via step-by-step mechanistic explanations formatted with '<think>' tokens. This transparency is crucial, allowing researchers to scrutinize the model's logic and translate computational outputs into testable scientific hypotheses.

The broader impact of this work lies in its potential to accelerate biological discovery. BIOREASON offers a robust tool for gaining deeper, mechanistic insights from genomic data, aiding in understanding complex disease pathways and the formulation of novel research questions. The development and application of benchmarks focused on multi-step reasoning, as utilized in this study, will further propel the advancement of AI systems capable of sophisticated biological understanding.

**Limitations.** Despite its strengths, BIOREASON has several limitations. Reliance on curated datasets such as KEGG introduces potential biases and limits coverage of less-characterized regions. The computational cost of encoding long DNA sequences and applying reinforcement learning (GRPO) raises training and inference time, reducing scalability to whole-genome or real-time settings. DNA sequences were truncated to 2048 tokens due to hardware limits, potentially omitting distal context. Finally, lack of robust uncertainty quantification limits reliability in high-stakes decisions.

**Future Work.** Future work will focus on expanding BIOREASON's scope and applicability. Key directions include incorporating orthologous sequences to enhance data diversity and model generalizability, and adapting the core framework to other biological modalities such as RNA and protein sequences, thereby addressing a broader range of research questions. Additionally, BIOREASON's improved variant effect prediction capabilities can be leveraged for impactful applications in genome-wide association studies (GWAS) and clinical mutation interpretation.

# 7 Conclusion

BIOREASON advances computational biology by seamlessly integrating high-capacity DNA sequence encoders with the flexible reasoning of large language models, yielding a unified framework that excels at both mechanistic pathway inference and variant pathogenicity prediction. Across KEGG-derived reasoning tasks and VEP benchmarks, our DNA–LLM hybrids consistently outperform models restricted to a single modality while generating transparent, stepwise explanations that facilitate expert validation. This tight multimodal fusion, further refined through reinforcement learning, not only boosts accuracy but also opens new avenues for interpretable genomic analysis. Future efforts will focus on scaling model size and data, designing leaner architectures, and leveraging modalities such as protein and RNA to broaden BIOREASON 's utility in medicine and biological discovery.

# Acknowledgments

We would like to thank Parsa Idehpour for his foundational contributions to the design and implementation of the multimodal GRPO infrastructure, which was essential for large-scale experimentation. We are deeply grateful to Guillaume Filion for his thoughtful guidance on the selection of biologically meaningful benchmarks. We also thank Ronald Xie for his valuable insights into model architecture. We appreciate Arman Sayed-Ahmadi for stimulating discussions that helped shape the project's future research directions. Finally, we extend our gratitude to Clem Delangue for his support and encouragement in making our models publicly accessible through the Hugging Face platform.

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

# A  Training Details

## A.1  Hyper-Parameters

All experiments share the following base settings unless otherwise noted. Please look at our GitHub for all other details and training scripts.

**Optimizer & regularization (SFT).**

- Optimizer: AdamW
- Learning rate: $5 \times 10^{-5}$
- Weight decay: $1 \times 10^{-2}$
- Gradient accumulation: 8 steps
- Random seed: 23
- Devices: 1

**LoRA adapters (SFT).**

- Rank: 32, Alpha: 64, Dropout: 0.05

**Optimizer & regularization (GRPO).**

- Optimizer: AdamW
- Learning rate: $1 \times 10^{-5}$
- Weight decay: $1 \times 10^{-2}$
- Gradient accumulation: 4 steps
- Learning Rate Scheduler: Cosine, 0.03 warmup ratio
- Random seed: 23

**LoRA adapters (GRPO).**

- Rank: 16, Alpha: 32, Dropout: 0.00

**GRPO Parameters.**

- Number of generations: 8
- Per device batch size: 8
- Steps: 1000 (7 epochs)
- Devices: 2
- Temperature (4B parameters): 0.7
- Temperature (1.7B parameters): 1
- Top p: 0.95
- Top k: 20
- Beta: 0.0
- Epsilon: 0.2

**DeepSpeed & hardware.**

- Strategy: `deepspeed_stage_2`
- CPUs per task: 8
- RAM per node: 128–256 GB
- Data-loader workers: 4

- **Task-specific settings:**
- *KEGG pathway reasoning:*
  - Batch size: 1
  - Epochs: 5
  - Max legnth DNA: 2048
  - Max text length: 1024 (for LLM only increases to 8192 to fit the raw DNA sequences)
- *Variant effect prediction (coding & non-SNV):*
  - Batch size: 2
  - Epochs: 3
  - Max legnth DNA: 2048
  - Max text length: 1024 (for LLM only increases to 8192 to fit the raw DNA sequences)

## A.2 Computational Resources

We conducted experiments using multiple GPU clusters equipped with NVIDIA A100 and H100 GPUs. A100 systems were equipped with Intel Xeon Silver CPUs, featuring 16-24 CPU cores, 24-32 threads, and 188-251 GB of RAM. We used 4 A100 GPUs for reinforcement learning, while other experiments were performed on single H100 GPUs with Slurm-based orchestration and Deepspeed.

## A.3 Reward Details

We use a deterministic composite reward function adapted from [7, 12] to guide reinforcement learning with GRPO, emphasizing correctness and strict adherence to the reasoning format. Each completion is parsed using an XML-aware extractor that isolates the final answer following the last `</think>` tag.

For each output $i$, the total reward $r_i$ is defined as the sum of the following components:

- **Correctness Reward.** Rewards $+2.0$ if the extracted final answer matches or contains the ground-truth answer (case-insensitive substring match), and 0.0 otherwise.

- **Conciseness Reward.** Rewards $+0.5$ if the extracted final answer contains ten or fewer words, encouraging brevity in final responses and preventing loopholes around the correctness reward.

- **Format Reward.** Rewards $+0.5$ if the completion strictly follows the required reasoning structure of a single `<think>` block followed by a newline and final answer, with properly closed tags.

Each reward component is computed independently and summed per sample, yielding a total reward in $[0, 2.5]$. Rewards are non-differentiable and used solely within GRPO to compute group-normalized advantages. No learned critic or value function is used.

## A.4 GRPO Details

Formally, given an input query $X_{\text{LLM}}$, GRPO samples a set of $G$ outputs $\{o_1, \ldots, o_G\}$ from the current policy $\pi_{\theta_{\text{old}}}$. Each candidate output $o_i$ comprises a reasoning trace and a final response. Outputs are evaluated using a composite domain-specific reward function $r(q.o_i)$, incorporating the rewards from Appendix A.3.

Dr. GRPO normalizes these rewards into an advantage using the average and standard deviation:
$$A_i = r_i - \text{mean}(\{r_1, \ldots, r_G\}) \tag{3}$$

The policy parameters $\theta$ are then optimized by maximizing the clipped surrogate objective:

$$\mathcal{J}_{\text{GRPO}}(\theta) = \mathbb{E}[X_{\text{LLM}} \sim P(Q), \{o_i\}_{i=1}^{G} \sim \pi_{\theta_{\text{old}}}(O|X_{\text{LLM}})]$$

$$\frac{1}{G} \sum_{i=1}^{G} \left( \min \left( \frac{\pi_\theta(o_i|X_{\text{LLM}})}{\pi_{\theta_{\text{old}}}(o_i|X_{\text{LLM}})} A_i, \text{clip} \left( \frac{\pi_\theta(o_i|X_{\text{LLM}})}{\pi_{\theta_{\text{old}}}(o_i|X_{\text{LLM}})}, 1-\epsilon, 1+\epsilon \right) A_i \right) - \beta \mathbb{D}_{\text{KL}}(\pi_\theta || \pi_{\text{ref}}) \right) \tag{4}$$

with hyperparameters $\epsilon$ and $\beta$.

## A.5 GRPO Learning Curve

Figure 4 illustrates the reward progression during GRPO training across three model configurations. Several key patterns emerge from the training dynamics:

**Rapid initial learning.** All models exhibit steep reward increases in the first 100-200 steps, demonstrating that GRPO efficiently guides the policy toward correct reasoning patterns early in training.

**Model size effects.** The 4B parameter models reach stable high reward values significantly faster than the 1.7B model, with the larger models stabilizing around step 400 compared to step 800 for the smaller model. The 4B models also demonstrate lower variance in the later training stages, suggesting that increased model capacity provides more robust optimization under GRPO.

**Near-optimal performance.** All models eventually reach rewards approaching the theoretical maximum of 2.5, indicating successful acquisition of both correct answer generation and adherence to the required reasoning format. The plateau behavior after stabilization suggests stable policy optimization without catastrophic forgetting.

**Architecture-agnostic learning.** Both NT (Nucleotide Transformer) and Evo2 DNA encoders paired with Qwen3-4B backbones follow nearly identical learning curves, indicating that the GRPO training procedure generalizes effectively across different DNA foundation model architectures.

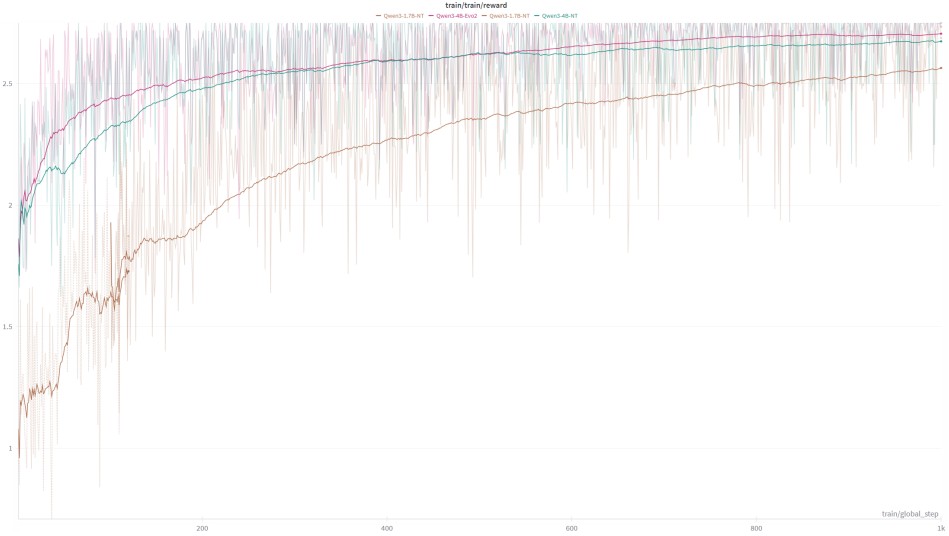

Figure 4: Mean reward progression during GRPO training across different BioReason model configurations over 1000 training steps. Shaded regions represent per-step reward variance across the batch of 8 generations per query.

# B  Generalization to Supervised DNA Models

To demonstrate that BIOREASON's architecture generalizes beyond unsupervised DNA foundation models, we conducted an additional experiment integrating a supervised DNA model, Enformer [3], for chromatin accessibility prediction. This experiment addresses two key questions:
(1) Can BIOREASON leverage supervised DNA encoders trained on specific genomic tasks?
(2) Does the framework extend to prediction tasks beyond variant-to-disease reasoning?

While our primary experiments utilized unsupervised DNA foundation models that learn general-purpose representations from diverse genomic data, supervised models like Enformer offer complementary strengths. Enformer is explicitly trained to predict genomic annotations including chromatin accessibility, outperforming DeepSEA on DNase-seq variant effect prediction [3]. By integrating Enformer into the BIOREASON framework, we evaluate whether our architecture can extract additional value from task-specific pretrained representations.

## B.1  Experimental Setup

We utilized a public dataset derived from the DeepSEA paper [44] containing 60,000 data points for long-range genomics. The human genome was segmented into 200bp bins, with each bin representing a classification instance. The task was formulated as multi-label binary classification to predict chromatin accessibility states ("open" vs "closed" for transcription factor binding) across 20 different DNase-seq tracks encompassing diverse cell types.

We chose to frame this as a classification task because DNase-seq data naturally represents binary chromatin accessibility states. While Enformer was originally trained for regression on epigenomic coverage prediction, our autoregressive architecture is optimized for discrete prediction tasks. Enformer's demonstrated competency with DNase-seq data, as shown in its original evaluation, suggests it retains relevant knowledge for this formulation.

## B.2  Model Configurations

We evaluated the following models:

- **Enformer (DNA-only):** Standalone Enformer with frozen weights and a trained attention head for classification.
- **Qwen3-1.7B and Qwen3-4B (LLM-only):** Qwen3 models fine-tuned on the task with DNA sequences treated as text.
- **Enformer-Qwen3-1.7B and Enformer-Qwen3-4B (DNA-LLM):** BIOREASON hybrids integrating Enformer embeddings with Qwen3 backbones.

All models were trained on a subset of the data and evaluated on a held-out test set following the experimental setup.

## B.3  Results

Table 3 presents comprehensive results across multiple evaluation metrics. The BIOREASON hybrids significantly outperform both standalone DNA and LLM baselines. The Enformer-Qwen3-4B model achieves a Macro F1-score of 33.70%, nearly doubling the performance of standalone Enformer (17.18%). This substantial improvement demonstrates that integrating sequence embeddings with an LLM via the BIOREASON architecture provides significant performance gains even when using supervised DNA encoders. The Enformer-Qwen3-1.7B configuration similarly outperforms both component models in isolation.

These results provide two critical insights into BIOREASON's design. First, the significant performance improvement with Enformer validates that the synergy between DNA sequence embeddings and LLM reasoning is a general architectural principle, not an artifact of using unsupervised models. This confirms the fundamental value of combining specialized sequence encoders with flexible LLM reasoning engines, regardless of the encoder's training paradigm.

Second, these results illuminate our rationale for prioritizing unsupervised DNA foundation models in the main experiments. While supervised models like Enformer demonstrate strong performance when

Table 3: Performance comparison of $f_{\text{DNA}}$-only, LLM-only, and DNA-LLM (BIOREASON) models on chromatin accessibility prediction across 20 DNase-seq tracks (all metrics in %). "M" and "W" denote *Macro* and *Weighted* averages, respectively.

| Model | F1-M | F1-W | Prec-M | Prec-W | Rec-M | Rec-W |
|---|---|---|---|---|---|---|
| **[DNA] Enformer** | 17.18 | 15.86 | 15.61 | 14.85 | 31.07 | 29.92 |
| **[LLM] Qwen3 - 1.7B** | 13.01 | 12.94 | 20.46 | 20.11 | 9.89 | 9.87 |
| **[LLM] Qwen3 - 4B** | 18.96 | 18.32 | 24.47 | 23.62 | 16.49 | 16.06 |
| **[DNA-LLM] Enformer + Qwen3 - 1.7B** | 25.89 | 24.39 | 27.02 | 25.55 | 33.97 | 32.45 |
| **[DNA-LLM] Enformer + Qwen3 - 4B** | **33.70** | **33.08** | **34.01** | **33.49** | **40.02** | **39.29** |

integrated into BIOREASON, they are highly specialized for their training objectives, in Enformer's case, predicting a predefined set of epigenetic marks. The standalone Enformer's modest performance (17.18% F1) on this task suggests that its specialized embeddings, while powerful, may not be as transferable to novel downstream tasks compared to the rich, general-purpose representations learned by unsupervised models from vast, diverse genomic data.

For BIOREASON's central goal of enabling broad, multi-step biological reasoning across diverse queries and tasks, unsupervised models provide a more robust foundation. Nevertheless, this experiment demonstrates that BIOREASON's architecture is flexible enough to accommodate both supervised and unsupervised DNA encoders, allowing users to select the most appropriate foundation model for their specific application domain.

## C  Per-Disease Performance on the KEGG-Derived Reasoning Benchmark

This section reports class-wise results for the KEGG-derived biological reasoning benchmark, providing a detailed view of BioReason's performance across individual disease categories. Each class corresponds to a distinct disease entity, with metrics averaged across multiple variant instances. The results demonstrate consistently strong performance across diverse diseases, indicating that BioReason generalizes well beyond high-frequency classes and maintains stable reasoning quality across varied mechanistic contexts.

Table 4: Per-disease performance on the KEGG-derived reasoning benchmark for NT + Qwen3-4B and Evo2 + Qwen3-4B (all metrics in %).

| Disease | Freq. | NT + Qwen3-4B | | | | Evo2 + Qwen3-4B | | | |
|---|---|---|---|---|---|---|---|---|---|
| | | Acc. | Prec. | Rec. | F1 | Acc. | Prec. | Rec. | F1 |
| **Parkinson's disease** | 47 | 99.7 | 97.9 | 100.0 | 98.9 | 100.0 | 100.0 | 100.0 | 100.0 |
| **Alzheimer's disease** | 40 | 99.7 | 100.0 | 97.5 | 98.7 | 100.0 | 100.0 | 100.0 | 100.0 |
| **Spinocerebellar ataxia** | 36 | 100.0 | 100.0 | 100.0 | 100.0 | 99.3 | 97.2 | 97.2 | 97.2 |
| **Amyotrophic lateral sclerosis** | 35 | 99.7 | 100.0 | 97.1 | 98.6 | 100.0 | 100.0 | 100.0 | 100.0 |
| **Melanoma** | 17 | 99.7 | 100.0 | 94.1 | 97.0 | 99.7 | 100.0 | 94.1 | 97.0 |
| **Prion disease** | 15 | 100.0 | 100.0 | 100.0 | 100.0 | 100.0 | 100.0 | 100.0 | 100.0 |
| **Colorectal cancer** | 12 | 99.3 | 85.7 | 100.0 | 92.3 | 99.7 | 92.3 | 100.0 | 96.0 |
| **Huntington's disease** | 10 | 100.0 | 100.0 | 100.0 | 100.0 | 100.0 | 100.0 | 100.0 | 100.0 |
| **Gaucher disease** | 7 | 100.0 | 100.0 | 100.0 | 100.0 | 100.0 | 100.0 | 100.0 | 100.0 |
| **Acute myeloid leukemia** | 7 | 100.0 | 100.0 | 100.0 | 100.0 | 100.0 | 100.0 | 100.0 | 100.0 |

