# OpenReview forum: "BioReason: Incentivizing Multimodal Biological Reasoning within a DNA-LLM Model"
_NeurIPS.cc/2025/Conference — NeurIPS 2025 poster_

### Official Review · Reviewer_ZvqQ · 2025-07-02

**Clarity:** 4
**Significance:** 4
**Originality:** 4
**Rating:** 6
**Confidence:** 4

**Summary:**

This paper presents BioReason, a model architecture that integrates a DNA foundation model with a LLM to improve prediction accuracy and enable natural language reasoning for each prediction. The training process combines supervised fine-tuning with reinforcement learning to encourage multi-step reasoning. As part of this work, the authors also developed and curated new benchmark datasets for biological reasoning. BioReason is evaluated using two different DNA foundation models and two LLMs across these datasets and shows marked accuracy improvements as well as useful step-by-step reasoning.

**Questions:**

1. Line 131: What are the effects of truncation?
2. Suggestion: The notation is clear with a detailed read, but including labels for major components (eg $\textbf{E}^\prime_{\text{DNA},k}$ vs $\textbf{E}^\prime_{\text{DNA}}$ in Figure 1 could help ease the comprehension, especially for the steps described in 3.3. “Stacking” along which dimension?
3. In section 4.2, what was the reasoning for the chosen data split by chromosome?
4. What was the performance like across different disease representations?
5. How do models of similar size (ie $\mathcal{f}\_{LargeLLM} \approx \text{BioReason}(\mathcal{f}_{SmallLLM}$ , $\mathcal{f}\_{DNA})$ compare? Could improved performance just be a factor of a larger joint model?
6. Suggestion: Reference Figure 3 in section 5.5, perhaps at the end of the first sentence.

**Ethical Concerns:**

["NO or VERY MINOR ethics concerns only"]

**Final Justification:**

This paper introduces BioReason, a modular framework that integrates DNA foundation models with LLMs for biological reasoning. The contributions are significant: curated benchmark datasets, evaluation across multiple backbones, and convincing performance towards explainable multimodal model. The paper is exceptionally well-written, clear, and accessible. My main concern was around the framing of interpretability. The authors clarified this in the rebuttal, explicitly defining their scope and limitations. They also provided thorough responses to all methodological questions and strengthened the case that performance gains stem from high-quality model performance rather than model size alone.

Overall, this is a technically strong, clear, and impactful paper. I maintain my Strong Accept (6) recommendation.

**Limitations:**

1. Since GPT was used in the dataset creation process, there is a non-negligible risk of errors or hallucinations in the generated data. This introduces potential noise that could affect model training or evaluation, and it would be valuable to include an error analysis or quality check to quantify this risk.
2. A user study could further explore the interpretability claims—specifically, by evaluating whether domain experts find BioReason’s outputs more trustworthy, understandable, or useful compared to a typical classifier. If experts tend to agree with BioReason’s explanations, it would also be important to examine how they respond when the model is wrong—do the explanations still seem convincing, and could they mislead users?

**Quality:**

3

**Strengths And Weaknesses:**

Strengths
- Clarity: This is a very well-written paper—clear, well-organized, and easy to follow. The motivations are explicitly laid out, arguments are well supported, and the structure flows naturally, almost telling a story. The writing is polished, concise, and approachable. Thank you for a very enjoyable read.
- Significance/Originality/Quality: The introduction of curated datasets and benchmarking for DNA LLM reasoning represents a valuable contribution to the field.
- Quality: The inclusion of stratification to ensure balanced disease representation is a thoughtful design choice.
- Quality/Significance: The modular design, in which $\mathcal{f}\_{\text{LLM}}$ and $\mathcal{f}\_{\text{DNA}}$ can adopt different backbones allows for flexibility in model size and the potential for performance improvements as new architectures emerge.
- Quality: The pairing of two SOTA $\mathcal{f}\_{\text{LLM}}$ and two SOTA $\mathcal{f}\_{\text{DNA}}$ models adds to the rigor to the evaluation, with results presented clearly using standard evaluation metrics.

Weakness:
- The term interpretability is overloaded in the field, so clearly defining it at the outset would help set expectations and clarify the nature of the claims. For instance, the approach does not offer inherent interpretability. Moreover, the LLM-generated explanations may not accurately reflect the true reasoning behind the model’s predictions—LLMs are known to produce plausible-sounding justifications for incorrect outputs and nonsensical rationales for correct ones. Without clearer framing at the outset, the scope and limitations of interpretability in this work only become apparent after a close and thorough reading of the paper.

---

> ### Author Rebuttal · Authors · 2025-07-31
>
> We thank the reviewer for their thorough review, detailed feedback, and especially the thoughtful suggestions! Below, we address the identified weaknesses.
>
> **Interpretability.** Thank you for raising this important point and suggestion. We will revise our writing in the form of a new subsection of the paper, touching on the scope and limitations of our framing of interpretability.
>
>
>
> **Questions.**
> 1. We appreciate this question about the effects of truncation. We used a 2048 nucleotide context window centered on each variant, following established practices from prior Nucleotide Transformer and Evo2 studies. As the context window was chosen after data curation, we truncated our sequences during batching to fit sequences to this 2048 nucleotide window.
>
>    We acknowledge that truncation may result in the loss of distal regulatory elements or long-range genomic context that could be relevant for variant interpretation. However, we have not systematically evaluated how increasing the context window would affect performance, as this was constrained by our computational resources and model limitations. This represents an important direction for future work - understanding the trade-offs between computational efficiency and the inclusion of longer-range genomic context.
>
>    We will clarify the truncation methodology in section 3.1 of the final manuscript and acknowledge this limitation in our discussion.
>
> 2. We thank the reviewer for pointing out this space for improved communication. We will update Figure 1 with labels matching the steps from 3.3, as well as specify that we concatenate the embeddings along the token dimension.
>
> 3. We utilized the established chromosome-based split strategy from the InstaDeepAI/genomics-long-range benchmark, which is a processed version of the data from the GPN-MSA paper. This splitting approach tests model generalizability to an entirely new genomic context and represents a rigorous evaluation strategy. We maintained this split in our dataset to ensure direct comparability with existing benchmarks and to provide a fair assessment of BioReason’s generalization capabilities across chromosomal regions.
>
> 4. Please see Tables 1-2 below for results.
>
> **Performance Across Disease Representations.**
> ### Table 1: NT + Qwen3 - 4B
>
> | # | Disease | Frequency | Accuracy | Precision | Recall | F1-Score |
> |---|---|---|---|---|---|---|
> | 0 | Parkinson’s disease | 47 | 0.997 | 0.979 | 1.000 | 0.989 |
> | 1 | Alzheimer’s disease | 40 | 0.997 | 1.000 | 0.975 | 0.987 |
> | 2 | Spinocerebellar ataxia | 36 | 1.000 | 1.000 | 1.000 | 1.000 |
> | 3 | Amyotrophic lateral sclerosis | 35 | 0.997 | 1.000 | 0.971 | 0.986 |
> | 4 | Melanoma | 17 | 0.997 | 1.000 | 0.941 | 0.970 |
> | 5 | Prion disease | 15 | 1.000 | 1.000 | 1.000 | 1.000 |
> | 6 | Colorectal cancer | 12 | 0.993 | 0.857 | 1.000 | 0.923 |
> | 7 | Huntington’s disease | 10 | 1.000 | 1.000 | 1.000 | 1.000 |
> | 8 | Gaucher disease | 7 | 1.000 | 1.000 | 1.000 | 1.000 |
> | 9 | Acute myeloid leukemia | 7 | 1.000 | 1.000 | 1.000 | 1.000 |
>
> ### Table 2: Evo2 + Qwen3 - 4B
>
> | # | Disease | Frequency | Accuracy | Precision | Recall | F1-Score |
> |---|---|---|---|---|---|---|
> | 0 | Parkinson’s disease | 47 | 1.000 | 1.000 | 1.000 | 1.000 |
> | 1 | Alzheimer’s disease | 40 | 1.000 | 1.000 | 1.000 | 1.000 |
> | 2 | Spinocerebellar ataxia | 36 | 0.993 | 0.972 | 0.972 | 0.972 |
> | 3 | Amyotrophic lateral sclerosis | 35 | 1.000 | 1.000 | 1.000 | 1.000 |
> | 4 | Melanoma | 17 | 0.997 | 1.000 | 0.941 | 0.970 |
> | 5 | Prion disease | 15 | 1.000 | 1.000 | 1.000 | 1.000 |
> | 6 | Colorectal cancer | 12 | 0.997 | 0.923 | 1.000 | 0.960 |
> | 7 | Huntington’s disease | 10 | 1.000 | 1.000 | 1.000 | 1.000 |
> | 8 | Gaucher disease | 7 | 1.000 | 1.000 | 1.000 | 1.000 |
> | 9 | Acute myeloid leukemia | 7 | 1.000 | 1.000 | 1.000 | 1.000 |
>
>
> 5. We thank the reviewer for this insightful question. Table 2 of the paper provides strong evidence that our performance gains are not simply due to increased model size, but due to a higher quality DNA embedding being passed into the LLM.
>
>    A particularly insightful comparison is between Qwen3-4B and NT + Qwen3-1B. The BioReason model (NT + Qwen3-1B) has fewer total parameters than the standalone Qwen3-4B, yet performs significantly better on variant effect prediction tasks. We believe this is due to the quality of the cross-modal embeddings from the DNA model, which is then integrated as part of the LLM’s prediction context. The LLM can then leverage these representations to achieve superior performance on biological tasks, even with fewer parameters. This suggests that BioReason's improvements result from the combination of domain-specific DNA representations and LLM reasoning capabilities.
>
> 6. We thank the reviewer for this suggestion to tie the discussion directly to the figure, and will make this change in the final manuscript.
>
>
>
>
>
>
>
> **Dataset Creation Hallucination Risk.**
> We acknowledge that this is an important concern. To mitigate this risk, we curated ground truth data on mutation effects and downstream phenotypic impacts from the KEGG database, which contains expert-curated summaries with citations on mechanistic effects. This served as our gold standard, and we used Claude solely to reformat these verified paragraphs into chain-of-thought reasoning traces rather than generating novel content.
>
> To assess quality, we conducted manual validation on a random sample of generated reasoning traces to verify biological accuracy and consistency. While quantifying exact confidence scores remains an open issue for LLM-generated content, our approach of providing comprehensive context and restricting the model's role to reformatting ground truth data mitigates hallucination risk.
>
> This methodology, combined with our validation sampling, provides confidence that our reasoning traces are biologically accurate and constitute reliable training data.
>
>
>
> **User Study.**
> We thank the reviewer for this excellent suggestion regarding user studies to evaluate our interpretability claims. We wholeheartedly agree that a comprehensive evaluation with domain experts would provide invaluable insights into BioReason's real-world utility and guide improvements for practical use with biologists studying unknown mutations.
>
> Such a study would require extensive coordination with domain experts, such as clinical geneticists. We believe this evaluation is important, but would be better suited as dedicated follow-up work that can properly address the complexities of interpretability in genomics.

---

> > ### Author Response · Authors · 2025-08-06
> >
> > Dear Reviewer,
> >
> > We hope this message finds you well. We wanted to follow up on our rebuttal and would be most grateful to learn whether our additional experiments and clarifications have adequately addressed your concerns. We deeply value your expert feedback and would be enormously appreciative if you could share your thoughts at your earliest convenience.
> >
> > Thank you again for your invaluable time and constructive review.

---

> ### Comment · Reviewer_ZvqQ · 2025-08-06
>
> Thank you, Authors, for the thorough rebuttal and additional details.
>
> \textbf{Interpretability} : What definition of interpretability will be used?
>
> \textbf{Question Rebuttals} :
> 1. Thanks for the elaboration and thoughtfulness about incorporating the explanation into the paper.
> 2 & 6. Already a well written paper, and looking forward to seeing the even clearer version. Thanks!
> 3. Great, thank you for the clarification and attention to methodological detail.
> 4. Thank you for the additional details - great to see fairly consistent performance across classes.
> 5. Convincing, thank you for pointing out the data that supports this strength of the model.
>
> \textbf{Hallucinations} : Thank you for your arguments.
> \textbf{User Studies }: Yes, looking forward to this dedicated follow up work.

---

> > ### Author Response · Authors · 2025-08-08
> >
> > Dear Reviewer,
> >
> > Thank you for your thoughtful feedback and for highlighting the importance of clearly defining interpretability. We define interpretability in our work as the model's ability to provide step-by-step natural language explanations that translate high-dimensional DNA sequence representations into human-readable reasoning chains connecting genomic mutations to disease outcomes. We acknowledge this represents post-hoc explainability rather than inherent interpretability, the explanations reflect learned associations between DNA embeddings and LLMs biological knowledge rather than true causal mechanisms, and we cannot guarantee the natural language reasoning accurately represents the foundation model's internal decision process. Despite these limitations, our approach provides valuable testable biological hypotheses, educational insights into potential genomic mechanisms, and a framework for experts to evaluate model reasoning. We view these explanations as hypotheses requiring biological validation rather than definitive mechanistic insights.

---

> > > ### Comment · Reviewer_ZvqQ · 2025-08-09
> > >
> > > Thank you, Authors, for clarifying the definition of interpretability used here. All my questions have been addressed, and I maintain my Strong Accept rating and confidence.

---

### Official Review · Reviewer_TdiE · 2025-07-03

**Clarity:** 3
**Significance:** 3
**Originality:** 2
**Rating:** 5
**Confidence:** 4

**Summary:**

This paper introduces BioReason, a DNA-LLM model that can reason about the effects of sequence mutations on downstream pathways and diseases. It merges two types of language models: genomic language models (such as EVO2 and NT) and large language models (such as Qwen3). Each types of these models are good for their own designed tasks. The gLMs are great at understanding DNA and its syntaxes and LLM are great for natural language. There is a disconnect between these two family of models, and BioReason tries to bridge the gap and make a native model that gets input from both DNA sequences and text query. The authors have benchmarked it on KEGG-derived biological reasoning datasets and variant effect prediction tasks and shown that it outperforms any other model (gLM and LLM) individually. This clearly shows a potential of merging these two types of models in order to boost biological reasoning.

**Questions:**

- Why have you only used unsupervised gLMs? I wonder what would be the performance of a supervised model like Borzoi and alphaGenome rather than EVO2.

**Ethical Concerns:**

["NO or VERY MINOR ethics concerns only"]

**Final Justification:**

The authors have addressed my concerns in the rebuttal.

**Limitations:**

Yes

**Quality:**

3

**Strengths And Weaknesses:**

Strengths:

- Merging gLMs and LLMs is an innovative and interesting idea to enhance reasoning.

- The benchmarks are convincing and clearly show the advantages of such hybrid models.

- The multi-step reasoning associating variants to downstream pathways and diseases is very helpful and has great use cases.

Weaknesses:

- The only gLMs used here is EVO and NT which are unsupervised models. It would be great to also show the performance of supervised models such as Enformer, Borzoi, and alphaGenome.

- Other than sequence to disease reasoning, it would be great to show how BioReason can handle sequence to gene expression and chromatin accessibility as well.

---

> ### Author Rebuttal · Authors · 2025-07-31
>
> We thank the reviewer for their positive assessment and insightful feedback. We are encouraged that they found our approach innovative and the results convincing. In response to the constructive weaknesses identified, we have conducted a new experiment that addresses both suggestions: evaluating our framework with a supervised DNA foundation model on a new task beyond sequence-to-disease reasoning.
>
>
> ---
>
>
> ### **New Experiment: Chromatin Accessibility Prediction**
>
> We evaluated the BioReason framework on a chromatin accessibility prediction task using the supervised DNA model, Enformer.
>
> * **Model Choice**: We specifically chose Enformer as it is a powerful, state-of-the-art supervised model explicitly trained to predict genomic annotations like chromatin accessibility, directly aligning with the reviewer's suggestions.
>
> * **Data and Task**: We utilized a public dataset for long-range genomics containing 60,000 data points. We trained all models on a subset of this data and evaluated their performance on a held-out test set. The task was formulated as a multi-label binary classification problem to predict chromatin accessibility across 20 different tracks.
>
>
>
> ### **Results**
>
> Our results demonstrate that the BioReason framework's core advantages hold true. The hybrid Enformer-Qwen models significantly outperformed both the standalone Enformer model and the Qwen LLMs.
>
>
> | Model | F1 Macro | F1 Weighted | Precision Macro | Precision Weighted | Recall Macro | Recall Weighted |
> | :--- | :--- | :--- | :--- | :--- | :--- | :--- |
> | Enformer | 0.1718 | 0.1586 | 0.1561 | 0.1485 | 0.3107 | 0.2992 |
> | Qwen3-1.7B | 0.1301 | 0.1294 | 0.2046 | 0.2011 | 0.0989 | 0.0987 |
> | Qwen3-4B | 0.1896 | 0.1832 | 0.2447 | 0.2362 | 0.1649 | 0.1606 |
> | Enformer-Qwen3-1.7B | 0.2589 | 0.2439 | 0.2702 | 0.2555 | 0.3397 | 0.3245 |
> | **Enformer-Qwen3-4B**| **0.3370**| **0.3308**| **0.3401**| **0.3349**| **0.4002**| **0.3929**|
>
>
> The Enformer-Qwen3-4B model doubled the F1 Macro score of the standalone Enformer model. This clearly shows that integrating sequence embeddings with an LLM via the BioReason architecture provides a substantial performance boost, even when using a supervised DNA encoder.
>
>
>
> ---
> ### **Discussion**
>
> This experiment provides two key insights:
>
> 1.  The BioReason framework is robust and generalizable. The significant performance lift demonstrates that the synergy between a DNA model and an LLM is a general principle, not an artifact of using unsupervised DNA encoders like Evo2 or Nucleotide Transformer (NT). This confirms the fundamental value of combining a specialized sequence encoder with the flexible reasoning engine of an LLM, regardless of the encoder’s training paradigm.
>
> 2.  Our initial choice of unsupervised DNA foundation models in the manuscript was deliberate. While supervised models like Enformer are powerful, they are highly specialized for the tasks they were trained on, such as predicting a predefined set of epigenetic marks. In contrast, unsupervised models like Evo2 and NT learn more general-purpose representations of genomic syntax and semantics by being trained on vast, diverse genomic data without being constrained to a specific set of labels.
>
> In our new experiment, the standalone Enformer model's performance was modest, suggesting its specialized embeddings, while potent, may not be as rich or transferable for novel downstream tasks. While our framework still extracts significant value, the overall performance ceiling is ultimately influenced by the quality and generalizability of the initial DNA embeddings. For our paper's central goal of creating a system for broad, multi-step biological reasoning, we believe the rich, flexible embeddings from unsupervised models provide a more robust foundation, better suited for the diverse and complex queries we aim for BioReason to handle.
>
>
> We thank the reviewer again for their valuable suggestions, which have helped us strengthen our work and further validate the robustness of the BioReason framework.

---

> > ### Author Response · Authors · 2025-08-06
> >
> > Dear Reviewer,
> >
> > We hope this message finds you well. We wanted to follow up on our rebuttal and would be most grateful to learn whether our additional experiments and clarifications have adequately addressed your concerns. We deeply value your expert feedback and would be enormously appreciative if you could share your thoughts at your earliest convenience.
> >
> > Thank you again for your invaluable time and constructive review.

---

> > > ### Comment · Reviewer_TdiE · 2025-08-06
> > >
> > > I would like to thank the authors to address my concern regarding the supervised models. I like that the new results show that BioReason can also work with the supervised models like Enformer. I will raise my score accordingly. In the meantime, the authors should also clarify in what cell types and tasks they are training Enformer, and why they choose to work with classification task, while Enformer has been trained to predict the coverage of the epigenomic tracks in a regression manner.

---

> > > > ### Author Response · Authors · 2025-08-08
> > > >
> > > > Dear Reviewer,
> > > >
> > > > Thank you for raising your score and for this important clarification question. You're absolutely right that Enformer was originally trained for regression on epigenomic coverage prediction. We chose to work with DNase-seq data because Enformer demonstrates competency with chromatin accessibility (as shown in Extended Figure 9 of the original Enformer paper, where it outperforms DeepSEA on DNase-seq variant effect prediction). Since our dataset derives from the DeepSEA paper, we reasoned that Enformer should have relevant knowledge for this task. We framed this as classification because DNase-seq data naturally represents binary chromatin accessibility states ("open" vs "closed" for transcription factor binding). While regression would be ideal, our autoregressive architecture cannot effectively perform regression task, a limitation we acknowledge for future work. Our dataset encompasses diverse cell types from the DeepSEA paper, with the human genome segmented into 200bp bins, where each bin represents a classification task predicting chromatin accessibility states across multiple tracks.

---

### Official Review · Reviewer_X1Q7 · 2025-07-05

**Clarity:** 3
**Significance:** 3
**Originality:** 3
**Rating:** 5
**Confidence:** 4

**Summary:**

The paper introduces BIOREASON, a groundbreaking multimodal AI framework that integrates a DNA foundation model with a large language model (LLM) to enable deep, interpretable biological reasoning. Unlike traditional DNA models that act as black boxes, BIOREASON allows the LLM to directly process genomic sequences and generate step-by-step, biologically coherent explanations.

Key highlights include:

1. A novel architecture that fuses genomic and textual data for multi-step reasoning.

2. Training with supervised fine-tuning and reinforcement learning to enhance logical deduction.

3. Superior performance on tasks like disease pathway prediction and variant effect analysis, with up to 15% improvement over single-modality baselines.

4. Transparent reasoning traces that explain predictions in a biologically meaningful way.

BIOREASON represents a major step forward in AI-driven biology, offering both accuracy and interpretability for complex genomic analysis.

**Questions:**

refer to Strengths And Weaknesses

**Ethical Concerns:**

["NO or VERY MINOR ethics concerns only"]

**Limitations:**

refer to Strengths And Weaknesses

**Paper Formatting Concerns:**

no formatting concern

**Quality:**

3

**Strengths And Weaknesses:**

Strengths
1. Multimodal Innovation: BIOREASON is the first architecture to deeply integrate a DNA foundation model with a large language model (LLM), enabling direct reasoning over genomic sequences.

2. Interpretable Reasoning: Unlike traditional “black-box” models, BIOREASON generates step-by-step, biologically coherent explanations using special tokens like <think>, enhancing transparency and scientific trust.

3. Strong Empirical Performance: Achieves up to 97.24% accuracy and 86.30% F1-score on KEGG-based reasoning tasks, outperforming both DNA-only and LLM-only baselines by significant margins.

4. Benchmark Contributions: Introduces new biological reasoning benchmarks, including a curated KEGG-derived dataset with mechanistic variant-to-disease pathways.

5. Reinforcement Learning Enhancement: Uses Group Relative Policy Optimization (GRPO) to further refine reasoning quality beyond supervised fine-tuning.

6. Case Study Validation: Demonstrates real-world utility by accurately predicting disease outcomes (e.g., ALS) with detailed mechanistic justifications.

7. Generalization Across Tasks: Performs well on both mechanistic reasoning and variant effect prediction, showing versatility across biological domains.

Weaknesses
1. Dataset Bias: Heavy reliance on curated datasets like KEGG may introduce biases and limit generalizability to under-characterized genomic regions.

2. Computational Overhead: Integrating long DNA sequences and applying reinforcement learning increases training and inference time, which may hinder scalability.

3. Lack of Uncertainty Quantification: The model does not currently provide confidence estimates, which is a limitation for clinical or high-stakes applications.

4. Implementation Complexity: The architecture and training pipeline (e.g., multimodal input fusion, GRPO) are technically complex and may be difficult to reproduce or deploy without significant resources.

5. Limited Modal Diversity: Currently focused on DNA; future extensions to RNA, proteins, or multi-omics data are needed to broaden its biological utility.

---

> ### Author Rebuttal · Authors · 2025-07-30
>
> We thank the reviewer for their thorough evaluation and insightful feedback on our work. We appreciate the recognition of our contributions in multimodal innovation, interpretable reasoning, and benchmark development. Below, we address the identified weaknesses.
>
>
>
> **Dataset Bias and Generalizability.**
> We selected the KEGG dataset, despite its bias towards well-studied pathways, because its expert-curated nature was essential for our primary goal of training a model to perform deep, multi-step reasoning on genomic sequence. This required a foundation of ground-truth causal chains linking variants to disease phenotypes. KEGG provided the ideal, high-quality data necessary to rigorously benchmark this novel capability, which is a core contribution of our work.
>
> To prove the framework generalizes beyond these curated examples, we trained and evaluated it on two large-scale Variant Effect Prediction (VEP) benchmarks. These tasks challenge the model with over 86,000 coding variants sourced from broader databases like ClinVar and gnomAD, covering diverse mutation types including SNVs and non-SNVs. Many of these variants are under-characterized compared to the KEGG examples. BioReason’s high accuracy across these benchmarks confirms its ability to generalize, effectively classifying a wide array of variants without relying on the detailed pathway information used during its reasoning training.
>
>
>
> **Computational Overhead.**
> We have made several deliberate design choices to ensure our framework remains efficient. During training, the large DNA foundation model is kept frozen, and we employ Low-Rank Adaptation (LoRA) for parameter-efficient fine-tuning of the language model, which significantly reduces the training burden. Additionally, we truncate input DNA sequences to a fixed context length during training to save memory. For inference, our publicly available framework is designed for compatibility with optimized inference servers like vLLM, which dramatically reduces latency and enhances scalability for practical applications through efficient batching and memory management. This also drives down the computational cost of GRPO stage, which is mostly bottlenecked by compute required for generations.
>
>
>
> **Lack of Uncertainty Quantification.**
> We thank the reviewer for highlighting this critical issue. The lack of robust uncertainty quantification is indeed a significant limitation and a well-recognized, field-wide challenge for LLMs. Current LLMs are fundamentally optimized to predict the next most probable sequence of tokens, which often results in confident-sounding, definitive outputs, even when the underlying information is ambiguous. Because BioReason leverages an LLM as its core reasoning engine, it naturally inherits this limitation. However, we designed BioReason's core mechanism of generating interpretable reasoning traces as a practical step towards addressing this trust deficit. While the model does not output a formal confidence score, it produces transparent, step-by-step biological explanations that allow domain experts to scrutinize the model's logic, evaluate its deductions, and ultimately build confidence in its conclusions in a way that is impossible with conventional black-box models.
>
>
>
> **Implementation Complexity and Reproducibility.**
> To ensure our results are fully reproducible, we will make all data, training code, and model checkpoints publicly available. For deployment, we are preparing our models for release on standard platforms like HuggingFace and have designed the framework to be compatible with high-throughput inference engines like vLLM, ensuring fast and efficient operation. Importantly, our approach is not limited to large-scale systems; we have demonstrated that the BioReason architecture provides significant performance gains even when using smaller 1.7B parameter models, making the benefits of our framework accessible across different computational budgets.
>
>
>
> **Limited Modal Diversity and Scope.**
> We thank the reviewer for this elegant future work suggestion. Our primary goal with this work was to establish BioReason as a foundational proof-of-concept. We focused on the deep integration of a DNA foundation model with a large language model to prove that our architecture could successfully unlock interpretable, multi-step reasoning from fundamental genomic sequences. This focused approach allowed us to rigorously validate this novel architectural paradigm. Adapting the core framework to other modalities, such as RNA and protein sequences, is a key priority. This extension is a central part of our upcoming BioReason2 model and in line with our vision of training multimodal biological LLMs that can reason across DNA, RNA, and Protein.

---

> > ### Author Response · Authors · 2025-08-06
> >
> > Dear Reviewer,
> >
> > We hope this message finds you well. We wanted to follow up on our rebuttal and would be most grateful to learn whether our additional experiments and clarifications have adequately addressed your concerns. We deeply value your expert feedback and would be enormously appreciative if you could share your thoughts at your earliest convenience.
> >
> > Thank you again for your invaluable time and constructive review.

---

> > ### Comment · Reviewer_X1Q7 · 2025-08-08
> >
> > the authors have addressed my concerns.

---

### Note · Authors · 2025-08-13

We sincerely thank the reviewers and Area Chair for their constructive engagement, thoughtful feedback, and recognition of the novelty and rigor of our work. During the discussion period, we addressed all identified weaknesses with new experiments, clarifications, and additional methodological detail.

### Addressing Reviewer Feedback

- **Generalizability beyond curated datasets:** We showed that BioReason performs strongly on two large-scale VEP benchmarks (over 86,000 variants from ClinVar and gnomAD) and on a chromatin accessibility prediction task using the supervised Enformer model. These results confirmed substantial performance gains and demonstrated that the architecture is not limited to KEGG-derived data.
- **Supervised vs. unsupervised DNA encoders:** Our Enformer–Qwen experiments confirmed BioReason’s strong performance with supervised encoders. We also explained why our main focus is on unsupervised models, which offer broader, label-agnostic genomic reasoning.
- **Additional biological tasks:** The Enformer experiment directly addressed suggestions to go beyond sequence-to-disease reasoning, showing applicability to gene regulation–related tasks.
- **Interpretability framing:** We defined interpretability in our work as post-hoc explainability through step-by-step natural language reasoning traces grounded in DNA embeddings, noting these should be viewed as hypothesis-generating rather than definitive proof.
- **Implementation clarity:** We clarified integration steps, truncation methodology, per-disease performance, chromosome-based data splits, and embedding concatenation strategies. Figures were updated for clarity. We will release all code, data, and checkpoints for reproducibility.
- **Computational efficiency:** We described our use of frozen DNA encoders, LoRA fine-tuning, context-length truncation, and vLLM-compatible inference for efficient training and deployment.

### Closing

We believe BioReason marks a significant step in integrating DNA foundation models with LLMs for biological reasoning. It delivers state-of-the-art accuracy while producing transparent, expert-scrutinizable outputs. Its flexibility across tasks, paired with our commitment to open release, makes it both a valuable research tool and a strong basis for future multimodal biological AI systems. We thank the Area Chair and reviewers for their time and consideration and look forward to sharing BioReason with the community.

---

### Decision · Program_Chairs · 2025-09-17

**Decision:**

Accept (poster)

**Comment:**

his paper presents BioReason, a DNA-LLM model for reasoning about the effects of sequence mutations on downstream pathways and diseases. The model integrates genomic language models, like EVO2 and NT, with large language models, like Qwen3. Evaluation was conducted on biological reasoning datasets derived from KEGG and on variant effect prediction tasks. Results indicate that the combined model performs better than either genomic or large language models used individually.